

# Genetic diversity of *Venturia inaequalis* isolates from the scabs in apple trees in Gansu Province, China, using AFLP markers

Zhaolong Lü[1,2], Nana Hui[2], Li Wang[2], Guo Zheng[2], Senshan Wang[1] and Jiping Li[1,2]

[1] College of Plant Protection, Gansu Agricultural University, Lanzhou, China
[2] Institute of Plant Protection, Gansu Academy of Agricultural Sciences, Lanzhou, China

## ABSTRACT

Apple scab is a serious disease that restricts the growth of cultivated apples. The objective of this study is to investigate the genetic variations and genetic structure of *Venturia inaequalis* in Gansu Province, China. 108 isolates of the pathogen *V. inaequalis* from the Jingning, Lingtai, Jingchuan, Xifeng, Ning and Maiji regions were collected, and their genetic diversity was analyzed using AFLP molecular marker technique. The results showed that genetic diversity was present among the isolates but was not statistically significant. Genetic distance values ranged from 0.0095 to 0.0762. Cluster analysis results showed that the 108 isolates could be divided into two clusters using a similarity coefficient of 0.69. A total of 104 isolates were contained in cluster I while four isolates were contained in Cluster II. From the AMOVA analysis, 98% of variations were observed within the same region, while 2% were observed across different regions. The analysis of population structure showed that 108 isolates had two common ancestors, with the Jingning isolates mainly being derived from the red ancestor. PCoA analysis showed that the Jingning isolates were independent to a certain extent. The different geographical location caused the genetic difference of the isolates. The genetic diversity of apple scab in Gansu Province is greatly aided by this work, which also offers a theoretical foundation for the use of molecular markers in assisted breeding to create novel resistant types.

## INTRODUCTION

The major disease known as apple scab, which is brought on by *Venturia inaequalis* ((Cooke) G. Winter), affects the apple industry and economic benefits. In severe cases, it will lead to a decline in apple yield, especially in colder apple-growing regions around the world (*Bowen et al., 2011*). Apple scab was discovered in China for the first time in Hebei Province in the 1920s and has since gradually spread to twelve provinces (*Huang & Zhang, 1997*). Apple scab pathogen *V. inaequalis* mainly damages apple leaves and fruits, but it can also infect petioles, flowers, sepals and pedicels. Furthermore, it can also infect other plants such as crabapples, loquats and hawthorn (*Jha, Thakur & Thakur, 2009*). Apple scab is so

Corresponding author
Jiping Li, gslijp@163.com

widespread in some areas that apple production losses may reach up to 70% (*MacHardy, 1996*).

Researchers in many countries and regions have studied the genetic diversity of *V. inaequalis* with different molecular tools (*Guérin et al., 2004*; *Mansoor et al., 2019*; *Gladieux et al., 2010*). Amplified fragment length polymorphism (AFLP) molecular marker technique was used to genotype isolates from the UK and China. Within China, there were no significant differences in relation to their geography or cultivar origin, but the populations of the four varieties in the UK were significantly different (*Xu et al., 2008*). This explains the genetic differences in *V. inaequalis* among different varieties of apple trees in the UK, but not in China. *V. inaequalis* isolates from two orchards in the UK were genotyped using AFLP molecular marker technique and simple sequence repeats (SSR). The results showed that the isolates of an individual cultivar within each orchard varied significantly, as did the isolates of an individual tree of the same cultivar in the same orchard. These results indicated that resistant varieties could be used for disease management in mixed orchards (*Xu et al., 2013*). In addition, the genetic diversity of *V. inaequalis* from 28 plantations on five continents was examined. The comes about appeared that 88% of genetic variety happened inside populaces, and it is related to widespread migration within regions (*Gladieux et al., 2008*). At the same time, by utilizing SSR to analyze the genetic diversity of *V. inaequalis* populaces within the UK and China, the results showed that the genetic relationship between both regions was relatively close. This was because most of the apple dwarf seedlings planted in Northwest China were from the EM system in the UK (*Li et al., 2021*).

In China, Gansu Province is located in the apple-producing area of the Loess Plateau, and its apple cultivation area and output rank among the top in the country. At the same time, apple scab occurred less. Be that as it may, in later a long time, apple scab has gotten to be commonplace in Gansu Province, driving to genuine misfortunes and posturing a noteworthy danger to the feasible improvement of the apple industry (*Hu et al., 2008*). AFLP molecular marker technique is a convenient and reliable molecular marker technology. This approach has not been used to explore genetic variations and genetic structure of *V. inaequalis* in Gansu Province.

This study collected and isolated apple scab samples from different apple-producing areas in Gansu and analyzed the genetic diversity of the *V. inaequalis* population using AFLP molecular marker technique. It gives a hypothetical premise for clarifying the genetic variety and populace structure of *V. inaequalis* in Gansu Province.

## MATERIAL AND METHODS

### Isolation of samples

Samples collected from Gansu Province between June to September 2021 as appeared in Table 1. Figure 1 gives the areas where tests were collected. The monoconidial strategy was utilized for sample isolation. The spore suspension was included into water agar (WA), and air-dried. Single spores were marked under the microscope after 24 h of incubation at 25 °C. A sterile needle was used to collect the marked areas on the WA, which were then

**Table 1  Number of samples and information regarding collection locations.**

| Samples | Numbers | District | Host | Longitude | Latitude | Times |
|---------|---------|----------|------|-----------|----------|-------|
| PLJN | 1∼21 | Jingning, Pingliang | Fuji | 105.36° E | 35.33° N | 2021.6.18 |
| PLLT | 22∼35 | Lingtai, Pingliang | Fuji | 107.33° E | 34.61° N | 2021.7.12 |
| QYNX | 36∼41 | Ning, Qingyang | Fuji | 108.11° E | 35.26° N | 2021.7.12 |
| TSMJ | 62∼84 | Maiji, Tianshui | Starkrimson | 106.33° E | 34.36° N | 2021.8.17 |
| PLJC | 85∼100 | Jingchuan, Pingliang | Fuji | 107.36° E | 35.22° N | 2021.7.13 |
| QYXF | 101∼108 | Xifeng, Qingyang | Fuji | 107.37° E | 35.51° N | 2021.7.13 |

**Notes.**
The isolated strains are numbered based on their region followed by a number, for example: PLJN-10, TSMJ-7, etc.

placed in potato dextrose agar (PDA) for incubation at 25 °C in dark environment. The isolate was purified to obtain the target isolate when after 40 d of culture.

## DNA extraction

The isolates were incubated for 40 days at 25 °C in dark environment. Approximately 100 mg of fungal mycelia was then collected and ground into powder after being frozen with liquid nitrogen. The Fungal Genomic DNA Extraction Kit (Omega Bio-tek, Sangon Company, Shanghai, China) was used to extract nucleic acid per the operating instructions of the kit, and was then stored at −20 °C.

## AFLP markers

The AFLP molecular marker technique was modified from *Vos et al. (1995)*. *Mse I* and *EcoR I* were used for double digestion. Approximately 500 ng of DNA, 5 U of *Mse I*, 5 U of *EcoR I* (Takara-Bio, Beijing, China) and 4 µl of H Buffer (Takara-Bio, Beijing, China) were used in a reaction. ddH$_2$O was added to obtain a final volume of 40 µl. The digestion protocol involved operation at 37 °C for 3 h followed by 80 °C for 20 min. The product of the double digestion was ligated to the adapters *EcoR I* (10 µM) and *Mse I* (10 µM) *via* a reaction containing: 40 µl of the double digestion product, 1 µl of (35 U) T4 DNA ligase (Takara-Bio, Beijing, China), 1 µl of 10×T4 DNA ligase Buffer (Takara-Bio, Beijing, China) and 1 µl of *EcoR I* and *Mse I* adapters. ddH$_2$O was used to attain a final volume of 50 µl. The protocol involved operation at 16 °C for 12 h followed by 80 °C for 20 min. Digestion-ligation products were visualized *via* patterning on 1.2% agarose gel. After ligation, the product was diluted using a 1:9 ratio with ultrapure water.

Pre-selective amplification was performed using 10 µl of Taq PCR MasterMix (Sangon Company, Shanghai, China), 2 µl of a pre-selective primer mixture *E*-TTC/*M*-TAA (10 µM) (*Fu et al., 2010b*), 5 µl of the ligated product and 8 µl of ddH$_2$O. Pre-selective amplification cycling parameters were 94 °C for 5 min followed by 31 cycles at 94 °C for 45 s, 45 °C for 40 s and an extension at 72 °C for 60 s, and 72 °C for 10 min and the product was then stored at −20 °C for further use. After pre-selective amplification, the PCR product was diluted using a 1:9 ratio with ultrapure water.

The 25 most common polymorphic combinations were chosen as molecular markers (*Fu et al., 2010b*) and selectively amplified. Selective amplification was performed using 10 µl of Taq PCR MasterMix (Sangon Company, Shanghai, China), 2 µl of the selective primer mixture *EcoR I* and *Mse I* (10 µM), 5 µl of the pre-selection amplification product and 8

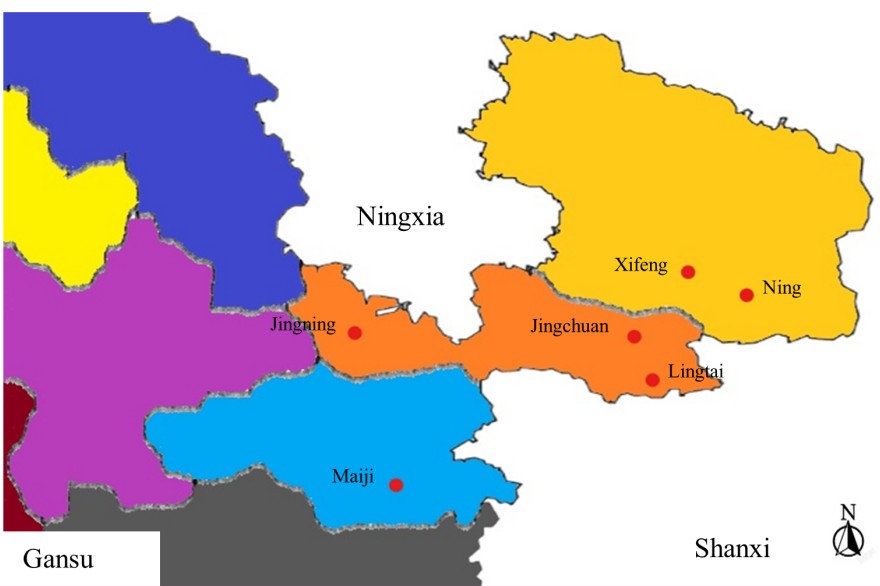

**Figure 1** **Areas from where sample collection was carried out.** Six regions in Gansu were chosen.

µl of ddH$_2$O. Selective amplification cycling parameters were 94 °C for 5 min followed by 15 cycles at 94 °C for 40 s and 65 °C for 30 s. This was followed by an extension at 72 °C for 60 s where the temperature decreased by 0.7 °C after each cycle. These cycling parameters were then followed by 30 cycles at 94 °C for 30 s, 51 °C for 30 s, and 72 °C for 60 s and a final extension at 72 °C for 10 min, followed by storage at −20 °C for further use.

## 6% Polyacrylamide gel electrophoresis

The SDS-PAGE Preparation kit (Sangon Company, Shanghai, China) was used for polyacrylamide gel electrophoresis. The electrophoresis buffer was prepared by diluting 10×TBE (100 ml) in 900 ml of distilled water. The electrophoresis program began with 40 min of pre-electrophoresis followed by 70 min of electrophoresis after sample loading. The sample loading volume was 10 µl. The gel silver staining method was modified from *Borzooeian et al. (2018)*. The method comprises of settling with 100 ml/L acetic acid (CH$_3$COOH) for 20 min, washing with ddH$_2$O for 1 min and silver staining with 1 g/L of silver nitrate solution (AgNO$_3$) for 20 min. The sample was then washed with ddH$_2$O for 10 s, developed using 16 g/L sodium hydroxide (NaOH), 8 ml/L formaldehyde (37%) solution for 5 min, a wash with ddH$_2$O for 10 s and a quenching reaction *via* the use of 7.5 g/L sodium carbonate (Na$_2$CO$_3$) for 5 min. The AFLP molecular marker bands were counted regardless of their presence or absence at the same position, with their presence marked as '1' and absence marked as '0'. A binary data '0-1' matrix was then established.

## Data analysis

Matrix data were imported to NTSYS2.1 to analyze genetic distances (*Weir & Ott, 1997*). Nei's genetic distance among the populations was estimated using POPGENE1.32 and GenAlEx 6.5 (*Kimura & Crow, 1964*; *Lewontin, 1972*; *Nei, 1973*; *Yeh et al., 1999*).

GenAlEx 6.5 was used for analysis of molecular variance (AMOVA) and analysis of principal co-ordinates (PCoA) (*Peakall & Smouse, 2006*). Individual genotypes were analyzed in Structure 2.3.4 (*Pritchard, Stephens & Donnelly, 2000*). The optimal K value was determined by uploading the analysis data to the platform Structure Harvester (http://taylor0.biology.ucla.edu/structureHarvester/) (*Earl & VonHoldt, 2012*).

## RESULTS

### Analysis of genetic diversity

Analysis of genetic diversity for *V. inaequalis* using 6% SDS-PAGE (Fig. 2). The results of Popgene1.32 are shown in Table 2. In the 108 isolates, the number of polymorphic loci (Np) was 301, the percentage of polymorphic loci (P) was 99.67%, and the effective number of alleles (Ne) was 1.4398. Nei's gene diversity (H) was 0.2712 and Shannon's information index (I) was 0.4241, which shown that there were a few genetic diversities within the populace of *V. inaequalis* in Gansu Province. The analysis results for *V. inaequalis* based on 6 geographic regions show that Np values ranged from 224 to 283, *P* values ranged from 74.17% to 93.17% while Ne values ranged from 1.3586 to 1.5506. H values ranged from 0.2206 to 0.3147 while I value ranged from 0.3432 to 0.4703. Based on the analysis of H and I, Jingning isolates had the highest level of genetic diversity, Maiji, Jingchuan, Xifeng, Lingtai isolates had the second highest level while Ning had the lowest level of genetic diversity.

### Cluster analysis

The unweighted pair group with arithmetic mean (UPGAM) method was used to cluster 108 isolates of *V. inaequalis* from 6 regions of Gansu Province. A tree graph (Fig. 3) was obtained, which was proven to be accurate *via* the cophenetic correlation test ($r = 0.8969$). As can be seen from Fig. 3, all the isolates tested were concentrated together when the similarity coefficient was 0.67. The isolates could be divided into two clusters when the similarity coefficient was 0.69. 104 isolates were contained in cluster I and 4 isolates were contained in Cluster II. Cluster I can be further divided into two sub-clusters, with Jingning isolates included in subcluster 1. Maiji, Lingtai, Jingchuan, Ning, Xifeng and several Jingning isolates were included in subcluster 2. Only Jingning isolates were included in Cluster II. The results show that there is no significant correlation between the isolates and their geographical locations in Gansu Province.

### Analysis of genetic distance

Popgene1.32 was used to analyze Nei's genetic identity and genetic distance (Table 3). These were obtained for six isolates from different geographical regions, in which isolates of Ning and Jingchuan have the highest level of genetic identity (0.9906) and the closest genetic distance (0.0095). This indicates that Ning and Jingchuan isolates have the smallest genetic difference. Xifeng and Jingning isolates have the lowest level of genetic identity (0.9267) and the furthest genetic distance (0.0762), which indicates that the genetic difference between Xifeng and Jingning isolates is the largest.

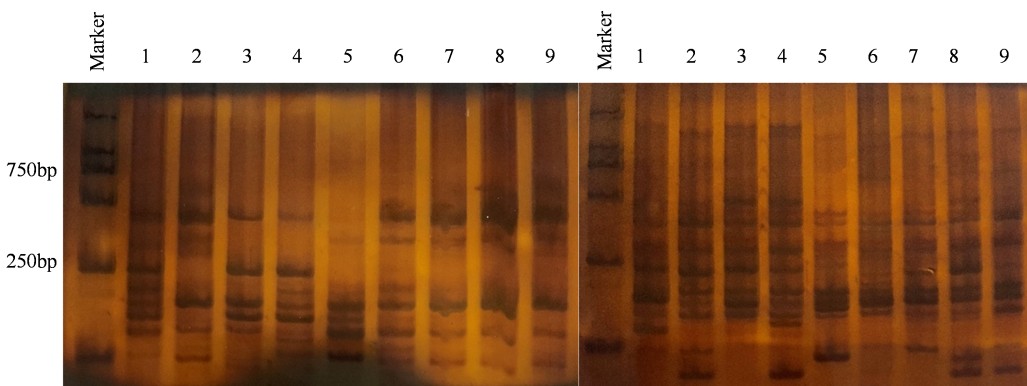

**Figure 2** **Result of partial isolates using 6% SDS-PAGE electrophoresis.** Primer combination E-CGC/M-GAC (left), primer combination E-CGC/M-CTG (right), Marker = 2000 bp, 1 - PLJN-3; 2 - PLJN-5; 3 - PLJN-13; 4 - TSMJ-4; 5 - TSMJ-7; 6 - TSMJ-15; 7 - PLJC-6; 8 - PLJN-15; 9 - QYXF-4.

**Table 2** **Analysis of genetic diversity for different populations of *V. inaequalis* in Gansu Province.**

| Code | N | Np | P (%) | Ne | H | I |
|---|---|---|---|---|---|---|
| Jingning | 21 | 283 | 93.71 | 1.5506 | 0.3147 | 0.4703 |
| Maiji | 23 | 224 | 74.17 | 1.3787 | 0.2268 | 0.3463 |
| Jingchuan | 16 | 238 | 78.81 | 1.3852 | 0.2323 | 0.3569 |
| Xifeng | 9 | 239 | 79.14 | 1.4553 | 0.2650 | 0.3985 |
| Lingtai | 13 | 242 | 80.13 | 1.4169 | 0.2476 | 0.3772 |
| Ning | 26 | 241 | 79.80 | 1.3586 | 0.2206 | 0.3432 |
| Total | 108 | 301 | 99.67 | 1.4398 | 0.2712 | 0.4241 |

**Notes.**

N, Number; Np, Number of polymorphic loci; P, Percentage of polymorphic loci; Ne, Effective number of alleles (*Weir & Ott, 1997*); H, Nei's gene diversity (*Kimura & Crow, 1964*); I, Shannon's information index (*Nei, 1973*).

**Table 3** **Nei's genetic identity and genetic distance among six different regions.**

| Populations | Jingning | Maiji | Jingchuan | Xifeng | Lingtai | Ning |
|---|---|---|---|---|---|---|
| Jingning | **** | 0.9387 | 0.9303 | 0.9267 | 0.9334 | 0.9353 |
| Maiji | 0.0632 | **** | 0.9808 | 0.9731 | 0.9725 | 0.9787 |
| Jingchuan | 0.0722 | 0.0194 | **** | 0.9893 | 0.9843 | 0.9906 |
| Xifeng | 0.0762 | 0.0273 | 0.0108 | **** | 0.9807 | 0.9818 |
| Lingtai | 0.0689 | 0.0278 | 0.0159 | 0.0195 | **** | 0.9876 |
| Ning | 0.0669 | 0.0216 | 0.0095 | 0.0184 | 0.0125 | **** |

**Notes.**

Nei's genetic identity (above diagonal) and genetic distance (below diagonal).

Asterisks (****) indicate no comparison of genetic differences was made between the same regions.

## Analysis of molecular variance (AMOVA)

AMOVA analysis of population variance (Table 4) showed that the coefficient of population differentiation (*PHiPt*) in the six geographical regions is 0.027 ($P < 0.05$). This indicates that there are some genetic variations for *V. inaequalis* in different regions of Gansu Province, but the genetic variations are not statistically significant (*PHiPt* <0.25). There is molecular variance in 108 isolates, among which the frequency of molecular variance is 98% within the same region and 2% across different regions. Therefore, it can be concluded

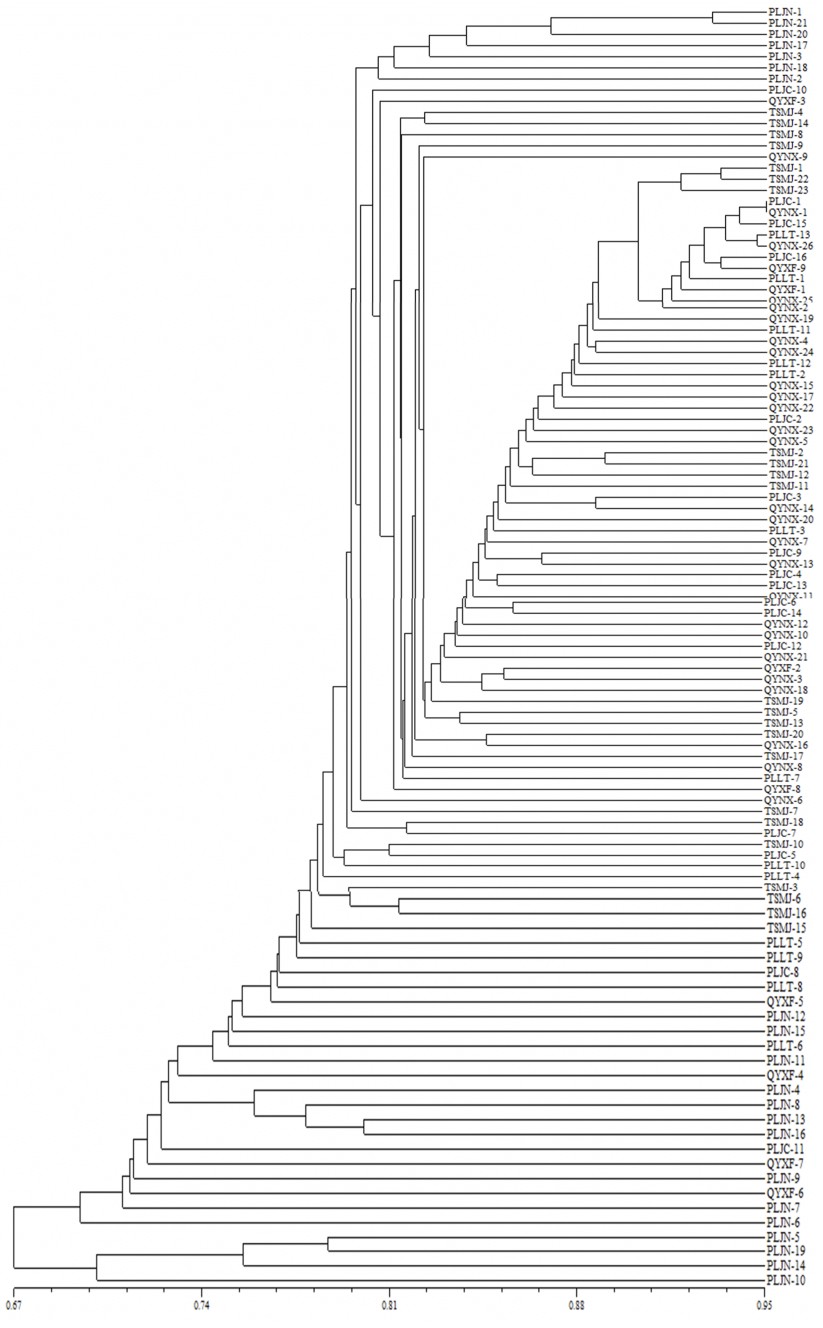

**Figure 3** Analysis of genetic relationship between 108 isolates of *V. inaequalis* based on 25 AFLP markers.

that there is greater molecular variance in the same region and less gene exchange between different regions.

The results of PCoA analysis (Fig. 4) showed that Jingning had more isolates in the fourth quadrant with several isolates independently distributed in the first quadrant. The genetic

**Table 4** AMOVA of *V. inaequalis* isolates grouped by geographic origin.

| Variation source | Degrees of freedom | Sum of aquares | MS | Estimate variance | Variance rate | *PHiPt* | *P* |
|---|---|---|---|---|---|---|---|
| Among Pops | 5 | 193.355 | 38.671 | 0.654 | 2% | 0.027 | 0.002 |
| Within Pops | 103 | 3375.000 | 15.617 | 31.250 | 98% | | |
| Total | 108 | 3568.355 | | 31.904 | 100% | | |

**Notes.**

MS, Mean-square error; *PHiPt*, Coefficient of population differentiation; p, Statistical significance.

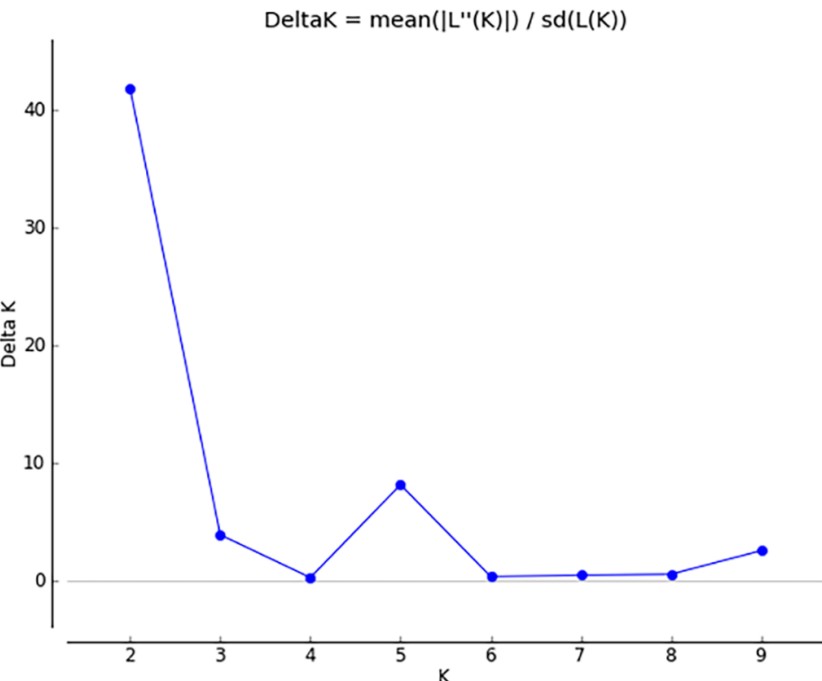

**Figure 4** Analysis of PCoA for *V. inaequalis* isolates sampled in Gansu.

structure of the tested isolates was analyzed, with results showing that the six regions in Gansu Province could be divided into 2 populations ($K = 2$, Fig. 5). As can be seen from Fig. 6, the genetic background of the Jingning isolates was mainly derived from ancestors indicated using a red color, with few green components. The genetic background of the other isolates involved different extents of contribution by both red and green ancestors with a high level of genetic similarity.

## DISCUSSION

Apple scab was to begin with recognized in Sweden in 1819 and has since spread to Germany, the United States, the United Kingdom and Australia. The disease was first discovered in China in Hebei Province and has now spread to 12 provinces such as Shaanxi Province, Gansu Province, Xinjiang Province and Henan Province (*Fu et al., 2010a*). In this places, large-scale epidemics have occurred with serious economic losses.

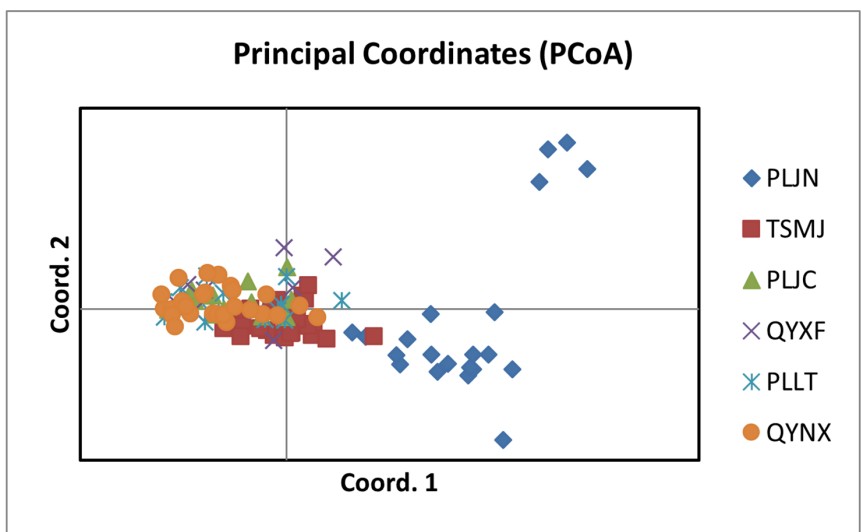

**Figure 5** Δ K peak value of 41.8 among assumed K-values.

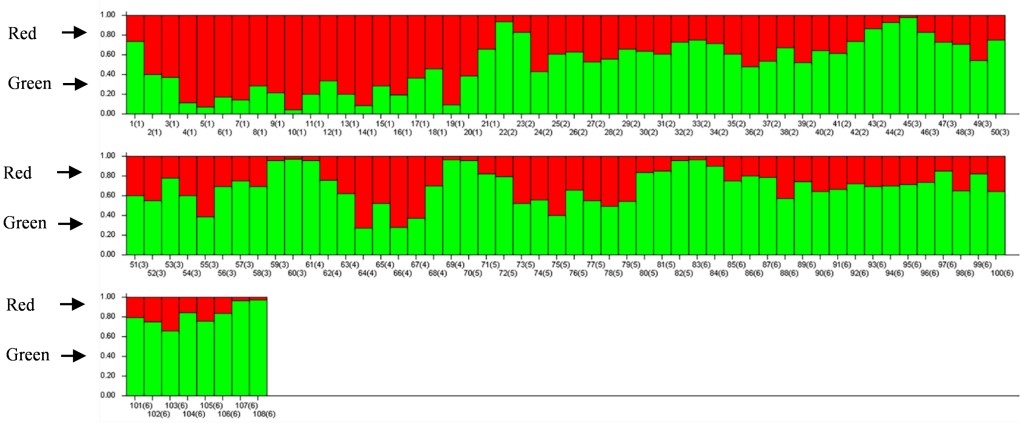

**Figure 6** **Analysis of *V. inaequalis* population structure shown using bar charts.** Above the dividing line is the red genetic background and below is the green genetic background.

AFLP is a molecular marker based on RFLP and PCR techniques created by Dutch scientists Zabeau and Vos. This method combines the simplicity of RFLP and RAPD and has the advantages of strong polymorphism, a simple methodology, the ability to handle a large amount of information and easy applicability (*Vos et al., 1995*). In this study, the AFLP molecular marker technique was used to analyze the genetic diversity of *V. inaequalis* from six different geographical regions of Gansu Province. The results showed that the populations of Jingning, Jingchuan, Lingtai, Maiji, Xifeng and Ning isolates were genetically similar and were consistent with the results of previous studies in China (*Hu et al., 2008*; *Fu et al., 2010a*; *Li et al., 2021*). The main environmental factors affecting the genetic variation of apples are the apple planting area and the pathogen hosts.

A higher level of genetic similarity across different geographical clusters may be due to the smaller sampling amount in the same province as well as smaller differences in climate and temperature (*Koopman et al., 2017*). The isolates of *V. inaequalis* in Gansu Province are not as affected by so many factors and are therefore unlikely to cause the pathogen to mutate. Furthermore, only *Fuji* and *Gala* apple varieties were involved in this sampling, so the role of different hosts in the genetic variation of *V. inaequalis* remains to be further explored.

AMOVA analysis showed that 98% of molecular variance frequency in the 108 isolates were within the same region and 2% were across different regions (Table 4). The results showed that the contribution of geographical location to population differentiation of *V. inaequalis* is small, which is similar to that of previous studies (*Ebrahimi et al., 2016*; *Koopman et al., 2017*; *Sitther et al., 2018*). It can also be seen that the genetic variation of apple scab was not mainly caused by its geographical location in Gansu Province. The author speculates that the frequent transportation of seedlings in Gansu Province may have further assimilated the genetic material of *V. inaequalis* and increased gene exchange within the population.

Using cluster analysis, 108 isolates were grouped into two clusters. In genetic structure analysis (Fig. 6), the isolates from six regions had two common ancestors, indicating a common genetic basis among the isolates (*Tenzer & Gessler, 1997*; *Tenzer & Gessler, 1999*; *Xu et al., 2008*). PCoA results showed that although most of the isolates were clustered together, the isolates from the Jingning area had some independence, with four isolates that were dispersed. These were PLJN-5, PLJN-10, PLJN-14 and PLJN-19 (Fig. 4). This may be due to the fact that Jingning is located to the west of the Liupan Mountains, with other areas being to the east of the Liupan Mountains. The presence of mountains makes it difficult for pathogens to spread geographically, contributing to a smaller gene flow. The large apple cultivation area of Jingning, which uses seedlings from all over the world has also increased the movement of pathogens between regions. Large-scale planting led to frequent exchanges of pathogenic isolates in the region and increased opportunities for new genetic variations. Therefore, some differences are observed in the clustering of *V. inaequalis* for Jingning. However, there were only four isolates with differences, so it is not possible to perform any further analysis.

At present, chemical fungicides are mainly used in the control of apple scab during apple cultivation. The long-term use of chemical fungicides will make pathogenic isolates resistant to drugs and aggravate environmental pollution. In addition, the use of different fungicides in different areas also has a significant influence on genetic variations of pathogenic isolates. Recently, new isolates that can overcome apple scab resistance in apple plants have been discovered (*Papp et al., 2020*; *Parisi et al., 1993*). In addition, several *V. inaequalis* isolates with resistance to multiple fungicides have been identified in the United States (*Lichtner et al., 2020*). All of these variables show genuine challenges to disease control. AFLP molecular marker examination of genetic diversity for *V. inaequalis* in Gansu Province was carried out in this study. If we want to know the genetic variation of *V. inaequalis* globally, further sampling investigations are necessary. Furthermore, this study used 6% SDS-PAGE electrophoresis to discover allele differential fragments. It is possible to use a

more efficient and convenient capillary electrophoresis (CE) method for analysis so as to improve efficiency and data accuracy (*Monnig & Kennedy, 1994*). This study provides a theoretical basis for explaining the genetic variation of *V. inaequalis* in Gansu Province.

### Funding

This project was supported by the Key R&D Program of Gansu Academy of Agricultural Sciences (2021GAAS07), the China Agriculture Research System on Apple Production (CARS-27) and the National Natural Science Foundation of China (31560487). The funders had no role in study design, data collection and analysis, decision to publish, or preparation of the manuscript.

### Grant Disclosures

The following grant information was disclosed by the authors:
Key R&D Program of Gansu Academy of Agricultural Sciences: 2021GAAS07.
China Agriculture Research System on Apple Production: CARS-27.
The National Natural Science Foundation of China: 31560487.

### Competing Interests

The authors declare there are no competing interests.

### Author Contributions

- Zhaolong Lü conceived and designed the experiments, performed the experiments, analyzed the data, prepared figures and/or tables, authored or reviewed drafts of the article, data curation, and approved the final draft.
- Nana Hui performed the experiments, prepared figures and/or tables, project administration, and approved the final draft.
- Li Wang performed the experiments, prepared figures and/or tables, and approved the final draft.
- Guo Zheng conceived and designed the experiments, authored or reviewed drafts of the article, project administration, and approved the final draft.
- Senshan Wang conceived and designed the experiments, analyzed the data, authored or reviewed drafts of the article, and approved the final draft.
- Jiping Li conceived and designed the experiments, authored or reviewed drafts of the article, project administration, and approved the final draft.

### Data Availability

The raw data is available at Zenodo: Zhao-long Lü. (2022). Peer J Raw Data. https://doi.org/10.5281/zenodo.7259955.

### Supplemental Information

Supplemental information for this article can be found online at http://dx.doi.org/10.7717/peerj.14512#supplemental-information.

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
