# Peer review of "Genetic diversity of Venturia inaequalis isolates from the scabs in apple trees in Gansu Province, China, using AFLP markers"

_PeerJ, doi:10.7717/peerj.14512_

## Round 0.1 · original submission · Minor Revisions

all reviewers comments should be adressed

Reviewer 1 ·

Basic reporting

well written manuscript

Experimental design

proper

Validity of the findings

proper

Additional comments

The manuscript of apple scab pathogen in Gansu province by markers to genetic diversity analysis, to explore the pathogenic bacteria in the genetic variation within the area and population genetic structure, provide theoretical basis for prevention and control of apple scab, innovative, clarity, content structure is reasonable, but English writing level remains to be further improved. Combined with the manuscript, the following suggestions are put forward for improvement:

1. Could the author explain how the sampling site was determined when the pathogen was collected?

2. Why were only 108 strains of pathogenic bacteria isolated from 267 samples of apple scab disease mentioned in the article?

3. The scientific name of pathogen appears for the first time, write the full name, the second time, the first word abbreviation, italics, please modify the full text.

4. How was the amount of enzyme used in the AFLP labeling process determined?

5. How is the reagent ratio determined during gel electrophoresis? Is there literature to support this?

·

Basic reporting

agree

Experimental design

agree

Validity of the findings

agree

Additional comments

This article through the analysis of the genetic diversity of apple scab pathogen, Gansu province, reveals its genetic background, molecular breeding for auxiliary apple provides certain theoretical foundation, innovative, clear, fluent, analysis, comprehensive, reliable data, but there are also some deficiencies, combined with the article puts forward the following Suggestions, hope to improve:
1. How was the PCR reaction procedure determined in the article? Is there literature to support this?
2. Please pay attention to the English writing format, check the whole article and revise it.
3. Why was the collection area of pathogenic bacteria determined as Gansu Province?
4. Whether the isolated pathogen is the target bacterium, please add in the materials, methods and results.
5. Why is polyacrylamide gel used instead of agarose gel for gel electrophoresis?

Reviewer 3 ·

Basic reporting

The manuscript provides certain theoretical basis, the article language is clear and fluent.Analysis is comprehensive, and methods are reproducible. However, the English writing level still needs to be further improved, and there are also some shortcomings. Combined with the article, I put forward the following suggestions for improvement:
1. Why was AFLP molecular marker method used in the experimental design? Why not use SSR, SNP and other methods?
2. Is apple Blackstar a major epidemic disease in Gansu Province? Please describe it in the introduction.
3. Please pay attention to the English writing format, check the whole article and revise it.
4. Whether the sampling area is representative, please describe.
5. Please add a description of the test process and method of AFLP molecular marker technology.

Experimental design

ok

Validity of the findings

ok

Additional comments

ok

---

## Round 0.2 · accepted · Accept

The authors have addressed all the reviewers' minor comments.